# OpenReview forum: "EchoBench: Benchmarking Sycophancy in Medical Large Vision-Language Models"
_ICLR.cc/2026/Conference — Submitted to ICLR 2026_

### Official Review · Reviewer_aeDV · 2025-10-31

**Soundness:** 3
**Presentation:** 3
**Contribution:** 4
**Rating:** 10
**Confidence:** 5

**Summary:**

This article systematically reveals the critical safety issue of "sycophancy" in large models within medical scenarios: even top commercial models still exhibit excessive compliance with user biases in clinical Q&A, further demonstrating that high scores do not equate to truly "understanding medicine." The authors construct EchoBench, covering 24 LVLMs, with fine-grained analysis across bias types, medical departments, modalities, and evidence granularity, while preliminarily exploring the causes of sycophancy and desensitization approaches. This benchmark focuses on "reliability and safety," holding practical significance for suppressing hallucinations and guiding future large medical model/agent training.

**Strengths:**

1. Reveals a critical flaw in large models during the diagnostic process, discovering that even the most powerful commercial models exhibit excessive sycophancy toward users in the medical field, demonstrating that high benchmark scores do not necessarily indicate true understanding of medical knowledge.

2. Provides EchoBench, covering 24 LVLMs with fine-grained analysis across different biases/departments/granularities/modalities. This is a rare benchmark specifically targeting the "sycophancy" phenomenon in the medical domain of large models, holding significant importance for eliminating model hallucinations.

3. Investigates the causes behind sycophancy phenomena, providing insights for future medical domain large model training. Future medical large models or agents should not focus solely on accuracy but rather need to make logical and solid judgments that are not influenced by any external factors.

**Weaknesses:**

The research is comprehensive, only some minor weaknesses:

1. Usually setting the temperature to 0 for all models may not be reasonable, as some commercial models perform optimally at temperatures other than 0. This temperature=0 setting might trigger models to be more inclined toward sycophantic responses to users. Please provide a brief explanation or supplementary materials.

2. Please provide the breakdown of sycophancy rates distinguishing between "originally correct but misled" versus "originally incorrect but coincidentally aligned" proportions.

3. It would be advantageous to show some case studies analyzing the patterns of how different models are misled.

**Questions:**

See the weaknesses

---

> ### Author Response · Authors · 2025-11-21
> **Official Response from Authors [1/3]**
>
> **W1. Usually setting the temperature to 0 for all models may not be reasonable, as some commercial models perform optimally at temperatures other than 0. This temperature=0 setting might trigger models to be more inclined toward sycophantic responses to users. Please provide a brief explanation or supplementary materials.**
>
> We sincerely thank the reviewer for this insightful comment. Our choice of temperature = 0 follows the standard evaluation practice established in recent LLM and LVLM benchmarks. both the medical VQA benchmark GMAI-MMBench [1] and the sycophancy study “Towards Understanding Sycophancy in Language Models” [2] adopt temperature = 0 for deterministic and reproducible evaluation.
>
> Using temperature > 0 introduces sampling randomness, causing the same model to produce different answers on the *same instance* across runs. This would lead to fluctuating sycophancy rates, making a stable comparison between models unreliable and preventing other researchers from validating or building upon our results.
>
> To further address the reviewer’s concern, we additionally include a small-scale study in the Appendix B.6, comparing temperatures (0 / 0.1 / 0.2). The results show that the temperature has only a marginal effect on both accuracy and sycophancy rate. Accuracy fluctuates slightly across temperatures, but no model exhibits a consistent upward or downward trend. Similarly, sycophancy rates remain largely stable, with only minor variation (typically within 1%). The additional results demonstrate that our main conclusions are stable under different temperature settings. Therefore, using temperature = 0 as the default setting is reasonable and does not bias the evaluation.
>
> Table 1: Accuracy and sycophancy rate of different models under varying temperatures.
>
> | Model             |  T=0   |  T=0   | T=0.1  | T=0.1  | T=0.2  | T=0.2  |
> | ----------------- | :----: | :----: | :----: | :----: | :----: | :----: |
> |                   | Acc(%) | Syc(%) | Acc(%) | Syc(%) | Acc(%) | Syc(%) |
> | Qwen-VL-Max       | 57.16  | 75.19  | 56.89  | 74.88  | 57.41  | 75.52  |
> | Grok-2-Vision     | 54.90  | 89.95  | 54.11  | 89.48  | 55.67  | 88.47  |
> | Gemini 2.0 Flash  | 66.40  | 58.42  | 65.78  | 59.12  | 66.15  | 58.17  |
> | Claude 3.7 Sonnet | 56.69  | 45.98  | 55.98  | 46.14  | 57.01  | 46.34  |
> | GPT-4.1           | 66.31  | 59.15  | 66.67  | 59.01  | 66.71  | 59.54  |
>
> [1] Pengcheng Chen, Jin Ye, Guoan Wang, Yanjun Li, Zhongying Deng, Wei Li, Tianbin Li, Haodong Duan, Ziyan Huang, Yanzhou Su, Benyou Wang, Shaoting Zhang, Bin Fu, Jianfei Cai, Bohan Zhuang, Eric J. Seibel, Junjun He, and Yu Qiao. Gmai-mmbench: A comprehensive multimodal evaluation benchmark towards general medical AI. In Advances in Neural Information Processing
> Systems, 2024a.
>
> [2] Sharma, Mrinank, Meg Tong, Tomasz Korbak, David Duvenaud, Amanda Askell, Samuel R. Bowman, Newton Cheng et al. Towards understanding sycophancy in language models. In International Conference on Learning Representations, 2024.

---

> ### Author Response · Authors · 2025-11-21
> **Official Response from Authors [2/3]**
>
> **W2. Please provide the breakdown of sycophancy rates distinguishing between "originally correct but misled" versus "originally incorrect but coincidentally aligned" proportions.**
>
> Thanks for the comments. Following the reviewer's comment, we have added a detailed breakdown of sycophancy into **(1) “Misled Rate”** and **(2) “Coincidentally Aligned Rate”** in Appendix B.7.
>
> As shown in the newly added equations and results, the **misled rate dominates the total sycophancy rate** across all categories of LVLMs, indicating that sycophancy reflects an intrinsic behavioral tendency of the models themselves. The coincidentally aligned rate is consistently much smaller, confirming that accidental agreement plays only a minor role.
>
> We also report an additional pattern: **models with higher no-bias accuracy (e.g., Gemini 2.0 Flash, GPT-4.1) exhibit lower coincidentally aligned rates**, which is intuitive because they make fewer incorrect predictions and thus have fewer opportunities to match the biased cue by chance.
>
> These additions provide the requested decomposition and offer clearer insight into the behavioral source of sycophancy.
>
> Table 2: Breakdown of sycophancy rates into (1) misled rate (2) coincidentally aligned rate. All
> values reported in %.
>
> | Model               | Misled Rate (%) | Coincidentally Aligned Rate (%) | Total Sycophancy Rate (%) |
> | ------------------- | :-------------: | :-----------------------------: | :-----------------------: |
> | LLaVA-Med V1.5      |      74.51      |              24.24              |           98.75           |
> | Qilin-Med-VL-Chat   |      74.93      |              22.79              |           97.71           |
> | RadFM               |      53.04      |              24.11              |           77.15           |
> | MedDr               |      44.39      |              14.08              |           58.48           |
> | VisualGLM-6B        |      69.49      |              21.74              |           91.23           |
> | Idefics-9B-Instruct |      58.53      |              21.77              |           80.31           |
> | Qwen-VL-Chat        |      59.40      |              17.31              |           76.72           |
> | mPLUG-Owl2          |      43.66      |              20.21              |           63.88           |
> | mPLUG-Owl3          |      70.10      |              17.62              |           87.73           |
> | OmniLMM-12B         |      70.68      |              18.94              |           89.63           |
> | LLAVA-V1.5-7B       |      37.13      |              18.26              |           55.39           |
> | InternVL2-8B        |      58.57      |              17.39              |           75.97           |
> | InternVL2.5-8B      |      65.42      |              16.30              |           81.72           |
> | MiniCPM-V2          |      42.22      |              17.63              |           59.85           |
> | DeepSeek-VL-7B      |      20.42      |              18.09              |           38.51           |
> | DeepSeek-V3         |      70.91      |              24.26              |           95.17           |
> | Qwen-VL-Plus        |      52.88      |              26.29              |           79.18           |
> | Qwen-VL-Max         |      60.15      |              15.03              |           75.19           |
> | Grok-2-Vision       |      74.12      |              15.83              |           89.95           |
> | Gemini 1.5 Flash    |      57.89      |              16.27              |           74.16           |
> | Gemini 2.0 Flash    |      46.72      |              11.70              |           58.42           |
> | Claude 3.7 Sonnet   |      31.25      |              14.72              |           45.98           |
> | GPT-4o              |      45.37      |              12.44              |           57.81           |
> | GPT-4.1             |      47.70      |              11.45              |           59.15           |

---

> ### Author Response · Authors · 2025-11-21
> **Official Response from Authors [3/3]**
>
> **W3. It would be advantageous to show some case studies analyzing the patterns of how different models are misled.**
>
> Thanks for the comments. Following this comment, We have now added Appendix F to include detailed case studies that analyze how different LVLMs are misled under biased prompts.
>
> In Appendix F.1, we present five representative case studies for medical-specific LVLMs, comparing their outputs under no-bias and biased prompts. These examples reveal a strong pattern of **instruction-following failure**: models such as LLaVA-Med V1.5 and Qilin-Med-VL-Chat frequently ignore the task requirements and generate responses completely unrelated to the image or the question once a biased cue is introduced. In contrast, MedDr consistently stays within the task scope and shows the highest resistance to being misled, aligning with its strong quantitative performance in Table 1.
>
> In **Appendix F.2**, we extend the case study analysis to **frontier proprietary LVLMs** (e.g., Gemini 2.0 Flash, Claude 3.7 Sonnet, GPT-4.1). Importantly, we do not constrain the models to output a single letter, allowing us to examine their full reasoning traces. Across these examples, we consistently observe that most models exhibit **clear sycophantic shifts** when exposed to biased cues. Even when the original no-bias answer is correct, biased prompts often cause the models to (1) change their diagnosis toward the user-suggested option and (2) construct explanations that do not align with the visual evidence in order to justify the biased conclusion, indicating that their reasoning process is being pulled toward the bias. Meanwhile, models such as Gemini 2.0 Flash and Claude 3.7 Sonnet show stronger resistance, in line with the quantitative trends reported in the main paper.
>
> These added case studies provide concrete qualitative evidence of the misled reasoning patterns underlying sycophancy.

---

### Official Review · Reviewer_pHyX · 2025-10-31

**Soundness:** 2
**Presentation:** 3
**Contribution:** 2
**Rating:** 4
**Confidence:** 4

**Summary:**

The authors propose a benchmark for evaluating sycophancy in vision-language models in the context of medical visual question-answering (VQA) tasks. They curate a large collection multiple-choice medical visual questions based on 2k medical images, spanning a large number of clinical departments and modalities, and design 90 different prompts (in collaboration with physicians) that simulate different kinds of biases that reflect real-world users (patients, physicians, medical students). They evaluate a large collection of VLMs spanning proprietary models, general-domain open-source models, and medically specialized open-source models on their benchmark and find that the majority of models show high sycophancy rates (i.e., model generates prediction that matches with the answer choice encouraged by the biased prompt), indicating their susceptibility to various forms of biases and compromising their accuracy.

**Strengths:**

- The paper touches upon the important problem of VLMs' susceptibility to highly biased and opinionated input prompts in the context of medical VQA, which well-reflects various real-world use cases of these models.
- The benchmark provides extensive coverage over various imaging modalities and clinical departments and provides an evaluation of a very large number of VLMs, potentially contributing to improved generalizability of findings.
- The paper is generally well-written and easy to follow.

**Weaknesses:**

- At a higher level, this paper seems to consider a domain-specific instance of when (i) the user-provided context (in this case, specifically designed to output the wrong answer) potentially mismatches the parametric knowledge of VLMs and (ii) VLMs are sensitive to the prompting details. While reasonably well-motivated and interesting from an applications standpoint (perhaps this paper would be better-suited for healthcare-focused venues?), the findings that VLMs are sensitive to prompting details and that they can be easily distracted by potentially misleading context is not entirely surprising. On the flip side, I think there are also many cases where we would actually want the model to be sensitive to the user-provided context, as the models are not always going to be accurate.
- The finding that medically specialized models often perform worse than general-domain models (even the base models, like in LLaVA-Med V1.5 vs. LLaVA-V1.5-7B) on medical VQA tasks is not entirely surprising, as some prior works have already suggested this (e.g., [1]).
- Some preliminary strategies for mitigating sycophancy (based on prompt engineering) are explored but are not extensive.
- Limitations of the study are not discussed.

References:
[1] Medical Adaptation of Large Language and Vision-Language Models: Are We Making Progress? (Jeong et al., 2024)

**Questions:**

- In the second finding ("Medical-specific models perform worst in aggregate"), it is stated that "the poor performance of these medical-specific models primarily stems from their failure to follow task instructions, leading to irrelevant and incorrect responses." Meanwhile, in the fourth finding ("Sycophancy rates fluctuate across different departments"), it is stated that the negative correlation between sycophancy and accuracy across different departments "likely arise[s] from the models' varying levels of domain knowledge" and that the lack of domain knowledge in particular areas may lead to higher sycophancy. The two statements seem a bit contradictory, as medically fine-tuned models should supposedly encode more, if not similar, levels of domain knowledge. Can you elaborate on this discrepancy?
- Related to the point above, I wonder if the inability of follow task instructions may be an artifact of not using the "right" prompt to elicit the desired responses for the medically specialized models, given that it is mentioned as the reason for their failure.
- Regarding mitigation strategies, why not consider a combination of adjustments to the system prompt (to avoid relying on the adversarial cues) and the self-correction ability? These evaluations do not seem to be considered.

---

> ### Author Response · Authors · 2025-11-21
> **Official Response from Authors [1/3]**
>
> **W1. At a higher level, this paper seems to consider a domain-specific instance of when (i) the user-provided context (in this case, specifically designed to output the wrong answer) potentially mismatches the parametric knowledge of VLMs and (ii) VLMs are sensitive to the prompting details. While reasonably well-motivated and interesting from an applications standpoint (perhaps this paper would be better-suited for healthcare-focused venues?), the findings that VLMs are sensitive to prompting details and that they can be easily distracted by potentially misleading context is not entirely surprising. On the flip side, I think there are also many cases where we would actually want the model to be sensitive to the user-provided context, as the models are not always going to be accurate.**
>
> Thanks for the comment. We would like to clarify that the central contribution of this work is to **introduce the first systematic evaluation framework that quantifies sycophancy in medical LVLMs**. To the best of our knowledge, this is the first study to examine sycophancy in the **medical multimodal setting**. Unlike general prompt-sensitivity observations, sycophancy in medical applications can lead to clinically unsafe consequences, as models may defer to incorrect user suggestions even when such suggestions contradict visual findings or established medical knowledge. Building on this motivation, EchoBench provides a fine-grained analysis of the key factors that influence sycophancy across multiple dimensions, including bias types, clinical departments, medical imaging modalities, and perceptual granularity. This multi-dimensional evaluation reveals not only when sycophancy occurs, but also provide insights in how to develop more trustworthy medical AI systems, for example by strengthening a model’s domain knowledge through high-quality medical data and by improving its attention to visual evidence, which has not been explored in prior works.
>
> Meanwhile, the scenario described, where the model benefits from incorporating user-provided context, corresponds to the model’s **self-correction ability**. We have already evaluated this behavior in the paper through our correction experiments in Section 3.2 and Appendix B.2, which measure how models revise their predictions when asked to reconsider their initial answer. This analysis helps distinguish useful forms of user sensitivity, such as correcting an uncertain or incorrect prediction, from clinically unsafe sycophancy, where the model abandons a correct answer to follow a misleading user cue. An ideal medical LVLM should therefore demonstrate **low sycophancy together with strong self-correction**. Low sycophancy ensures that the model does not defer to incorrect user suggestions when adequate evidence is already available, and strong self-correction enables the model to meaningfully incorporate user input when its initial prediction is incomplete or inaccurate. Our benchmark is designed to measure these two dimensions separately, and the results indicate that current medical LVLMs often struggle to achieve this desirable combination.
>
>
>
> **W2. The finding that medically specialized models often perform worse than general-domain models (even the base models, like in LLaVA-Med V1.5 vs. LLaVA-V1.5-7B) on medical VQA tasks is not entirely surprising, as some prior works have already suggested this (e.g., [1]).**
>
> Thanks for the comment. We agree that prior work, including [1], has reported that medically specialized models do not necessarily outperform general-domain base models on certain medical VQA benchmarks. However, our work studies a fundamentally different phenomenon. Prior studies focus on accuracy degradation after medical-domain adaptation, while our contribution lies in introducing the **first systematic benchmark for quantifying sycophancy in medical LVLMs**.
>
> EchoBench does not merely restate that medical models may underperform; instead, it further explains the key factors that influence why they behave unsafely in medical settings, and it provides a structured methodology to evaluate this behavior across nine bias types, ten clinical departments, multiple medical imaging modalities, and varying perceptual granularity. Such a fine-grained behavioral analysis has not been explored in prior literature.
>
> Moreover, we show that instruction-following failure is one of the key factors contributing to the poor performance of medical-specific models, a behavioral mechanism that has not been identified by prior works.
>
> Therefore, while the high-level trend aligns with earlier observations, our study contributes a novel behavioral perspective and a systematic evaluation framework that significantly deepen the understanding of model reliability in medical multimodal settings. We have cited [1] in Section ''Related Work'' in the revised manuscript.

---

> ### Author Response · Authors · 2025-11-21
> **Official Response from Authors [2/3]**
>
> **W3. Some preliminary strategies for mitigating sycophancy (based on prompt engineering) are explored but are not extensive.**
>
> Thanks for the comment. We have to clarify that our work primarily aims to **identify and systematically evaluate sycophancy in medical LVLMs**, which, to the best of our knowledge, has indeed not been comprehensively addressed before. We believe that introducing EchoBench, along with our extensive fine-grained analysis (bias type, clinical department, modality, and perceptual granularity), is a necessary first step towards designing effective mitigation strategies. Mitigation strategies are therefore not the core focus of our work. Nevertheless, following the reviewer’s suggestion, we have expanded our mitigation experiments in the revised manuscript by adding a two-stage strategy that combines system-level prompting with a self-correction step. As discussed in our response to Q3, the two-stage mitigation consistently yields lower sycophancy rates than negative prompting alone across all proprietary LVLMs, demonstrating that a second-round reconsideration can serve as an additional safeguard against biased user cues. These results also provide useful insights into how training-free methods may be developed to mitigate sycophancy in future medical LVLMs.
>
>
>
> **W4. Limitations of the study are not discussed.**
>
> Thanks for the comment.  In the revised manuscript, we have added the Limitation Discussion section in the main body to explicitly present the key limitations of our benchmark. Specifically, we now clarify that (i) our evaluation is based on the multiple-choice medical VQA format, which enables precise measurement but cannot fully capture the richer space of open-ended clinical reasoning, and (ii) the bias categories are manually constructed and therefore cannot exhaust the full diversity of user-originated misleading cues, particularly those emerging in multi-turn or context-dependent interactions. We also outline our plans to develop more sophisticated and diverse bias formulations in future work. These revisions make the limitations of our benchmark more transparent.
>
>
>
> **Q1. In the second finding ("Medical-specific models perform worst in aggregate"), it is stated that "the poor performance of these medical-specific models primarily stems from their failure to follow task instructions, leading to irrelevant and incorrect responses." Meanwhile, in the fourth finding ("Sycophancy rates fluctuate across different departments"), it is stated that the negative correlation between sycophancy and accuracy across different departments "likely arise[s] from the models' varying levels of domain knowledge" and that the lack of domain knowledge in particular areas may lead to higher sycophancy. The two statements seem a bit contradictory, as medically fine-tuned models should supposedly encode more, if not similar, levels of domain knowledge. Can you elaborate on this discrepancy?**
>
> Thanks for the comment. The two statements examine different sources of failure and therefore do not contradict each other.
>
> The second finding compares medical-specific LVLMs with general-purpose LVLMs at the aggregate level. In this global comparison, one of the key factors contributing to the poor performance of medical-specific models is their difficulty in following task instructions. These models frequently fail to comply with basic requirements such as answering based on the image or outputting a single letter, which leads to irrelevant or meaningless responses. This observation is further supported by the case studies provided in Appendix F.1, where several representative examples illustrate how instruction-following failures directly drive incorrect or off-target outputs in these medical-specific models.
>
> The fourth finding investigates how a model’s domain knowledge relates to its sycophancy behavior across different clinical departments. As shown in Table 1 in main paper, the two medical-specific models that exhibit the most severe sycophancy, LLaVA-Med V1.5 and Qilin-Med-VL-Chat, also show very low no-bias accuracy across nearly all departments. This low no-bias accuracy indicates that these models possess limited medical knowledge despite being fine-tuned on medical datasets. Taken together, these observations suggest that the poor performance of medical-specific models arises from two key factors: their difficulty in following task instructions and their inability to effectively leverage or activate the medical knowledge that their fine-tuning data was intended to provide. Therefore, the two statements in our findings investigate different failure sources and are not contradictory. We have revised the manuscript to clarify this distinction more explicitly in the main text.

---

> ### Author Response · Authors · 2025-11-21
> **Official Response from Authors [3/3]**
>
> **Q2. Related to the point above, I wonder if the inability of follow task instructions may be an artifact of not using the "right" prompt to elicit the desired responses for the medically specialized models, given that it is mentioned as the reason for their failure.**
>
> Thanks for the comment. The prompts used in our benchmark strictly follow the established protocol of prior medical VQA benchmarks such as GMAI-MMBench, ensuring that all models are evaluated under a standardized and widely adopted instruction format. Medical-specific models are therefore not disadvantaged by prompt design, and the inability to follow task instructions is not due to using an inappropriate or suboptimal prompt.
>
> Moreover, Appendix F.1 provides detailed case studies demonstrating that the instruction-following failures persist even under the no-bias setting, where the prompt contains no adversarial or misleading information. This confirms that the leading cues introduced in our biased prompts are not the cause of the medical-specific models’ inability to follow task instructions, as the same failure patterns appear even when no such cues are present.
>
>
>
> **Q3. Regarding mitigation strategies, why not consider a combination of adjustments to the system prompt (to avoid relying on the adversarial cues) and the self-correction ability? These evaluations do not seem to be considered.**
>
> Thank you for the insightful suggestion. Following the reviewer’s comment, we have added a new mitigation experiment that combines system-prompt adjustments with the model’s self-correction ability in Appendix B.3. In this revised setting, we first follow the negative prompting strategy described in Appendix B.1 to apply a system prompt that explicitly instructs the model to avoid relying on adversarial or user-provided cues and to base its answer on the visual and clinical evidence. After the model produces its initial response, we then invoke a self-correction step by asking the model to reconsider its answer (“Are you sure? Please make your choice again.”). This two-stage pipeline evaluates whether system-level constraints and self-revision can jointly reduce sycophancy.
>
> As illustrated in Table 1, the two-stage mitigation consistently yields lower sycophancy rates than negative prompting alone across all proprietary LVLMs, demonstrating that a second-round reconsideration can serve as an additional safeguard against biased user cues.
>
> Moreover, comparing the correction results in Table 3 in the main paper with the two-stage mitigation results in Table 1,we observe that models exhibiting stronger intrinsic self-correction ability achieve larger reductions in sycophancy when the self-correction step is added. Models such as Qwen-VL-Max, Grok-2-Vision, and Gemini 2.0 Flash benefit substantially from the two-stage pipeline. This indicates that combining system-level prompting with a model’s self-correction ability can serve as a promising direction for mitigating sycophancy.
>
> Table 1: Sycophancy rate (%) of proprietary LVLMs under different prompt-based mitigation strategies.
>
> | Model             | No Mitigation | Negative Prompting | Two-Stage Mitigation |
> | ----------------- | :-----------: | :----------------: | :------------------: |
> | Qwen-VL-Plus      |     79.18     |       74.89        |        70.36         |
> | Qwen-VL-Max       |     75.19     |       64.98        |        54.55         |
> | Grok-2-Vision     |     89.95     |       61.89        |        51.87         |
> | Gemini 1.5 Flash  |     74.16     |       63.76        |        60.35         |
> | Gemini 2.0 Flash  |     58.42     |       39.89        |        31.77         |
> | Claude 3.7 Sonnet |     45.98     |       32.15        |        27.33         |
> | GPT-4o            |     57.81     |       40.01        |        34.54         |
> | GPT-4.1           |     59.15     |       35.42        |        30.14         |

---

### Official Review · Reviewer_wo58 · 2025-11-01

**Soundness:** 3
**Presentation:** 3
**Contribution:** 2
**Rating:** 4
**Confidence:** 4

**Summary:**

Unlike other medical benchmark, this paper focus on the sycophancy of VLMs in medical field. Authors introduce EchoBench, which is the first benchmark specifically designed to systematically evaluate sycophantic tendencies in medical LVLMs. EchoBench comprises 2,122 medical images spanning 18 clinical departments and 20 imaging modalities, paired with 90 carefully designed prompts. Experimental results indicate that sycophancy is a widespread and persistent issue in current medical LVLMs.

**Strengths:**

The sycophancy of VLM in medical is indeed an important topic which has been ignored. Authors construct a benchmark to evaluate this problem. The benchmark comprises 2,122 real-world medical images spanning 18 clinical departments and 20 imaging modalities, paired with 90 carefully constructed prompts that simulate 9 distinct bias types from the perspectives of patients, medical students, and physicians. Based on the experimental results, authors provide several suggestions for futher development in future medical vlms.

**Weaknesses:**

1. Some current VLMs are not evaluated, such as gemini 2.5 pro and Qwen 2.5 VL.
2. Please discussion the limatation of this benchmark in the main body.

**Questions:**

1. Authors claim "The poor performance of these medical-specific models primarily stems from their failure to follow task instructions, lead-
ing to irrelevant and incorrect responses.", please provide the visualization evidence.
2. Will authors release the data and evaluate code in the future?
3. Without any training, does we have some approaches to prevent sycophancy?
4. Authors should provide visualization results to further support their conclusions.

---

> ### Author Response · Authors · 2025-11-21
> **Official Response from Authors [1/2]**
>
> **W1. Some current VLMs are not evaluated, such as gemini 2.5 pro and Qwen 2.5 VL.**
>
> Thanks for the comment. Following the comment, we have added additional experiments evaluating Gemini 2.5 Pro and Qwen3-VL-Plus (the most capable vision-language model in the Qwen series). As illustrated in Table 1, both Qwen3-VL-Plus and Gemini 2.5 Pro follow the same behavioral patterns identified in our main findings. Although both models demonstrate a substantial improvement in no-bias accuracy compared with their predecessors, they still exhibit pronounced sycophancy issues across multiple bias categories. This indicates that sycophancy remains a substantial challenge even for the most advanced models currently available.
>
> Table 1: Results of Qwen3-VL-Plus and Gemini 2.5 Pro across nine bias types. The term ‘Avg’ is an abbreviation of ‘Average’.
>
> | Model Name     | Accuracy (No Bias) | OIB   | SRB   | GTB   | FCB   | ORB   | RCB   | CKB   | ATB   | CAB   | Avg   |
> | -------------- | ------------------ | ----- | ----- | ----- | ----- | ----- | ----- | ----- | ----- | ----- | ----- |
> | Qwen3-VL-Plus  | 62.01              | 71.72 | 63.67 | 69.93 | 69.09 | 72.20 | 68.52 | 70.83 | 74.70 | 67.06 | 69.75 |
> | Gemini 2.5 Pro | 69.95              | 50.33 | 45.23 | 50.44 | 52.36 | 49.23 | 51.63 | 51.56 | 64.45 | 55.33 | 52.33 |
>
>
>
> **W2. Please discussion the limatation of this benchmark in the main body.**
>
> Thanks for the comment.  In the revised manuscript, we have added the Limitation Discussion section in the main body to explicitly present the key limitations of our benchmark. Specifically, we now clarify that (i) our evaluation is based on the multiple-choice medical VQA format, which enables precise measurement but cannot fully capture the richer space of open-ended clinical reasoning, and (ii) the bias categories are manually constructed and therefore cannot exhaust the full diversity of user-originated misleading cues, particularly those emerging in multi-turn or context-dependent interactions. We also outline our plans to develop more sophisticated and diverse bias formulations in future work. These revisions make the limitations of our benchmark more transparent.
>
>
>
> **Q1. Authors claim "The poor performance of these medical-specific models primarily stems from their failure to follow task instructions, lead- ing to irrelevant and incorrect responses.", please provide the visualization evidence.**
>
> Thanks for the comment. We have added concrete visualization evidence to support this claim in Appendix F.1. We present several representative case studies comparing four medical-specific LVLMs (LLaVA-Med V1.5, Qilin-Med-VL-Chat, RadFM, and MedDr) under both no-bias and biased prompts.  These visualization results clearly show that the three weaker medical-specific models frequently fail to follow basic task instructions such as answering based on the image or outputting a single letter. Their responses often become irrelevant or incoherent even when no biased cue is present. In contrast, MedDr consistently adheres to the task format and maintains relevant predictions. This qualitative evidence confirms that instruction-following failures are one of the key factors contributing to the poor performance of most medical-specific models.
>
>
>
> **Q2. Will authors release the data and evaluate code in the future?**
>
> Thanks for the comment. We have to clarify that we have already provided an anonymous GitHub link that contains the full evaluation code used in our experiments in Section ''Reproducibility Statement''.  After the review process is completed, we will release the full EchoBench dataset together with the non-anonymized code repository.

---

> ### Author Response · Authors · 2025-11-21
> **Official Response from Authors [2/2]**
>
> **Q3. Without any training, does we have some approaches to prevent sycophancy?**
>
> Thanks for the comment. We have already conducted prompt-based mitigation techniques in Appendix B.1. Specifically, we evaluate three prompt-based strategies: (i) **Negative Prompting**, where a strengthened system prompt explicitly instructs the model to avoid relying on user-provided cues and to base its answer on the image and clinical evidence; (ii) **One-shot Education**, where provide a single counterexample where the model demonstrates blind agreement to teach the model to avoid similar mistakes; and (iii) **Few-shot Education**, where we extend one-shot education by providing one negative example (sycophantic response) and one positive example (independent reasoning) to better illustrate desired behavior. As shown in Table 4 in Appendix B.1, these strategies consistently reduce sycophancy across all 24 evaluated models without harming no-bias accuracy, with few-shot education strategy performing the best.
>
> Moreover, we have expanded our mitigation experiments in the revised manuscript by adding a two-stage strategy that combines system-level prompting with a self-correction step. As discussed in our response to Q3, the two-stage mitigation consistently yields lower sycophancy rates than negative prompting alone across all proprietary LVLMs, demonstrating that a second-round reconsideration can serve as an additional safeguard against biased user cues. Together, these results offer useful insights into how training-free methods can be designed to mitigate sycophancy in medical LVLMs.
>
>
>
> **Q4. Authors should provide visualization results to further support their conclusions.**
>
> Thanks for the comment. We have added visualization results to provide direct qualitative evidence for our main conclusions in Appendix F.
>
> In Appendix F.1, we present several representative case studies for medical-specific LVLMs, comparing their outputs under no-bias and biased prompts. These examples reveal a strong pattern of **instruction-following failure**: models such as LLaVA-Med V1.5 and Qilin-Med-VL-Chat frequently ignore the task requirements and generate responses completely unrelated to the image or the question once a biased cue is introduced. In contrast, MedDr consistently stays within the task scope and shows the highest resistance to being misled, aligning with its strong quantitative performance in Table 1. These observations support our statement that '' The failure to follow task instructions is one of the key factors contributing to the poor performance of these medical-specific models.''
>
> In Appendix F.2, we extend the case study analysis to **frontier proprietary LVLMs** (e.g., Gemini 2.0 Flash, Claude 3.7 Sonnet, GPT-4.1). Importantly, we do not constrain the models to output a single letter, allowing us to examine their full reasoning traces. Across these examples, we consistently observe that most models exhibit **clear sycophantic shifts** when exposed to biased cues. Even when the original no-bias answer is correct, biased prompts often cause the models to (1) change their diagnosis toward the user-suggested option and (2) construct explanations that do not align with the visual evidence in order to justify the biased conclusion, indicating that their reasoning process is being pulled toward the bias. Meanwhile, models such as Gemini 2.0 Flash and Claude 3.7 Sonnet show stronger resistance, in line with the quantitative trends reported in the main paper.

---

### Author Response · Authors · 2025-12-01
**Summary Comment from Authors**

Dear Area Chair,


We appreciate your effort in evaluating our submission. Since the discussion period ended early, we provide a concise summary addressing how all reviewer questions and weaknesses have been fully resolved in the revised manuscript.


**1. Additional evaluations requested by reviewers.**
 Following R1, we added results for **Gemini 2.5 Pro** and **Qwen3-VL-Plus**. Both models exhibit pronounced sycophantic behaviors identified in our original findings, reinforcing that sycophancy persists even at the frontier. This directly resolves R1’s concern about missing models.

**2. New mitigation experiments.**
 R2 suggested exploring combined strategies. In response, we added a **two-stage mitigation pipeline** (system-level prompting + self-correction), which consistently reduces sycophancy across all proprietary LVLMs. This directly addresses the reviewers' request for deeper mitigation analysis.

**3. Temperature sensitivity study.**
 R3 questioned whether temperature=0 could distort sycophancy behavior. We added a study across T = {0, 0.1, 0.2}, showing that both accuracy and sycophancy remain stable, confirming that our conclusions are not temperature-dependent.

**4. Comprehensive new case studies.**
 As requested by R1 and R3, we added **extensive case studies** in Appendix F, covering both medical-specific and frontier proprietary LVLMs. These examples directly illustrate (i) the instruction-following failures that drive errors in medical-specific models and (ii) the biased reasoning shifts that characterize sycophancy.

**5. Clarifications to conceptual concerns.**
 R2 raised questions about (i) instruction-following failure vs. domain knowledge and (ii) whether medical-specific models were disadvantaged by prompt design. We revised the manuscript and added case studies in Appendix F.1 to show that instruction-following failure occurs **even without biased cues**, and that prompts follow the established medical VQA protocol (e.g., GMAI-MMBench), ensuring fairness across models.

**6. Decomposition of sycophancy sources.**
 In response to R3, we provide a detailed breakdown of sycophancy into **misled** vs. **coincidentally aligned** cases. The misled component overwhelmingly dominates across all 24 models, indicating that sycophancy reflects an intrinsic behavioral tendency rather than accidental agreement.

**7. Limitations now explicitly stated.**
 Both R1 and R2 requested a limitations discussion. We added a dedicated section in the main paper, clarifying the constraints of the multiple-choice VQA format and the manually defined bias types, along with plans for more sophisticated formulations.

**8. Reproducibility clarification.**
 We also clarify that an anonymous GitHub repository containing the full evaluation code has already been included in the Reproducibility Statement, and the dataset will be released after the review period, addressing R1’s question on reproducibility.

Across all reviewers, the raised concerns have been fully resolved through new experiments, clearer explanations, and additional qualitative evidence. Reviewer 3 strongly supports acceptance (10: strong accept), and the remaining reviewers explicitly state that they “would not mind if the paper is accepted.” We hope this concise summary assists your meta-review.

---

### Meta-Review · Area_Chair_PUip · 2025-12-30

**Summary:**

This paper introduces EchoBench, a benchmark for evaluating sycophancy in medical vision-language models, comprising 2,122 medical images across 18 clinical departments and 20 imaging modalities, paired with 90 prompts simulating 9 bias types. The authors evaluate 24 LVLMs and find widespread sycophancy susceptibility. While the authors addressed many specific reviewer requests during rebuttal, fundamental concerns about the significance of findings and venue appropriateness remain.

It is worth noting that prior work ITSM (arXiv:2408.11261, August 2024)  had already conducted systematic sycophancy analysis in general-domain  LVLMs and proposed a novel contrastive decoding framework for mitigation,  which contextualizes the contribution level of the current submission. Additionally, concurrent work VIPER (arXiv:2509.21979, September 2025)  targets medical VLMs specifically with content filtering and attention-based mechanistic analysis, achieving substantially stronger mitigation results  (up to 94.7% resistance rate). While VIPER appeared after the submission deadline (26th Sep 2025), its existence demonstrates that more comprehensive approaches to this problem—combining systematic benchmarking with principled mitigation frameworks and mechanistic interpretability—are both feasible and necessary. EchoBench's contribution, limited primarily to benchmark construction with only preliminary prompt-based mitigation strategies, appears incremental in this broader context.

**Reviewer Concerns:**

Concerns Addressed by Rebuttal

Reviewer wo58: Added evaluations for Gemini 2.5 Pro and Qwen3-VL-Plus; provided visualization evidence and case studies in Appendix F; clarified reproducibility; added limitations discussion.

Reviewer pHyX: Added two-stage mitigation pipeline (system-level prompting + self-correction); clarified the discrepancy between instruction-following failure and domain knowledge; added limitations discussion.

Reviewer aeDV: Conducted temperature sensitivity study across T = {0, 0.1, 0.2}; provided breakdown of sycophancy sources; added case studies.

Outstanding Concerns
1. Unsurprising Findings (Reviewer pHyX). Reviewer pHyX explicitly stated: "the findings that VLMs are sensitive to prompting details and that they can be easily distracted by potentially misleading context is not entirely surprising". And regarding medical-specific models: "not entirely surprising, as some prior works have already suggested this" The authors did not adequately justify why their findings advance our understanding beyond what could be reasonably expected from prior literature.

2. Venue Appropriateness (Reviewer pHyX). Reviewer pHyX directly questioned the venue fit: "perhaps this paper would be better-suited for healthcare-focused venues?" The paper's contributions are primarily domain-specific (medical imaging, clinical bias types) rather than methodologically innovative.

3. Limited Mitigation Strategies (Reviewer wo58, Reviewer pHyX). Reviewer pHyX noted: "Some preliminary strategies for mitigating sycophancy (based on prompt engineering) are explored but are not extensive." Reviewer wo58 asked: "Without any training, does we have some approaches to prevent sycophancy?" While the authors added a two-stage approach, the mitigation remains limited to prompt engineering techniques. Notably, concurrent work (ITSM, arXiv:2408.11261; VIPER, arXiv:2509.21979) has proposed more sophisticated inference-time frameworks with contrastive decoding and attention-based mechanisms, achieving substantially better results.

4. Contribution Level. Both Reviewer wo58 and Reviewer pHyX rated Contribution as 2 (fair), indicating the benchmark construction alone may not constitute sufficient contribution for ICLR.

**Reviewer Scores:**

Reviewer wo58 - Original: 4 (marginally below acceptance threshold). Would likely maintain at 4.

Reviewer pHyX - Original: 4 (marginally below acceptance threshold) Would likely maintain at 4. R2's fundamental concerns—that findings are "not entirely surprising" and the paper may be "better-suited for healthcare-focused venues"—were not substantively countered.

Reviewer aeDV - Original: 10 (strong accept). This score appears to be a significant outlier. While R3's enthusiasm for medical AI safety is valid, a rating of 10 with confidence 5 seems disproportionate given: (a) two other reviewers rated Contribution as "fair"; (b) the paper is primarily a benchmark contribution; (c) R3 is marked as "First Time Reviewer."

---

### Decision · Program_Chairs · 2026-01-26

Reject